# Biomolecular Evaluation of *Lavandula stoechas* L. for Nootropic Activity

**DOI:** 10.3390/plants10061259

**Published:** 2021-06-21

**Authors:** Aamir Mushtaq, Rukhsana Anwar, Umar Farooq Gohar, Mobasher Ahmad, Romina Alina Marc (Vlaic), Crina Carmen Mureşan, Marius Irimie, Elena Bobescu

**Affiliations:** 1Department of Pharmacology, Punjab University College of Pharmacy, University of the Punjab, Lahore 54000, Pakistan; aamir_mushtaq@hotmail.com (A.M.); rukhsanaanwar2003@yahoo.com (R.A.); ahmadmobasher@hotmail.com (M.A.); 2Gulab Devi Institute of Pharmacy, Gulab Devi Educational Complex, Lahore 54000, Pakistan; 3Institute of Industrial Biotechnology, GC University, Lahore 54000, Pakistan; dr.mufgohar@gcu.edu.pk; 4Food Engineering Department, Faculty of Food Science and Technology, University of Agricultural Sciences and Veterinary Medicine, 400372 Cluj-Napoca, Romania; crina.muresan@usamvcluj.ro; 5Faculty of Medicine, Transilvania University of Brasov, 500036 Brasov, Romania; elena.bobescu@gmail.com

**Keywords:** phenethylamine, *L. stoechas*, acetylcholine, choline acetyltransferase, AChE, aromatic amine, enhancement of memory

## Abstract

*Lavandula Stoechas* L. is widely known for its pharmacological properties. This study was performed to identify its biomolecules, which are responsible for enhancement of memory. *L. stoechas* aqueous extract was first purified by liquid column chromatography. The purified fractions were analyzed for in vitro anti-cholinesterase activity. The fraction that produced the best anti-cholinesterase activity was named an active fraction of *L. stoechas* (AfL.s). This was then subjected to GC–MS for identifications of biomolecules present in it. GC–MS indicated the presence of phenethylamine and α-tocopherol in AfL.s. Different doses of AfL.s were orally administered (for seven days) to scopolamine-induced hyper-amnesic albino mice and then behavioral studies were performed on mice for two days. After that, animals were sacrificed and their brains were isolated to perform the biochemical assay. Results of behavioral studies indicated that AfL.s improved the inflexion ratio in mice, which indicated improvement in retention behavior. Similarly, AfL.s significantly (*p* < 0.001) reduced acetylcholinesterase and malondialdehyde contents of mice brain, but on the other hand, it improved the level of choline acetyltransferase, catalase, superoxide dismutase, and glutathione. It was found that that high doses of AfL.s (≥400 mg/Kg/p.o.) produced hyper-activity, hyperstimulation, ataxia, seizures, and ultimate death in mice. Its LD_50_ was calculated as 325 mg/Kg/p.o. The study concludes that α-tocopherol and phenethylamine (a primary amine) present in *L. stoechas* enhance memory in animal models.

## 1. Introduction

*Lavandula stoechas* L. (Lamiaceae) is an aromatic and medicinal plant of the Mediterranean region. It was the most popular folk remedy for the management of digestive disorders, kidney disease, diabetes mellitus, hyperlipidemia, cough, asthma, headache, and flu-like symptoms [1,2]. It is also known as “broom of the brain” due to its extensive use in the treatment of migraines, epilepsy, and memory related disorders [3,4]. The plant is rich in camphor, erythrodiol, eucalyptol, fenchone, *lavanol*, longipene-2-ene, longipene-2-ene monoacetate, lupeol, luteolin, myrtenol, oleanolic acid, pinocarvyl acetate, terpineol, ursolic acid, vergatic acid, vitexin, α-amyrin, β-sitosterol, and a variety of aromatic compounds [5,6]. Linalool present in *L. stoechas* has a sedative effect [7] and it acts on different brain receptors to modify behavioral patterns, along with creating a sense of calmness and wellbeing. Camphor, present in *L. stoechas,* has stimulating effects on the brain [8]. Lavender oils extracted from *L. stoechas* are famous for their spicy fragrances and are used in aromatherapy to treat anxiety, depression, sleeplessness, headaches, and migraine [9]. Moreover, *L. stoechas* showed antispasmodic [8], sedative, anti-epileptic, antibacterial [10], antifungal [11], anti-leishmaniasis [5], anti-inflammatory [12], cytotoxic [13] and anti-diabetic [14] properties.

Keeping in mind the traditional uses of *L. stoechas*, as a neurotonic and memory enhancer, as part of our studies on the methanolic extract of *L. stoechas* [15], our current study was designed to report the active ingredient of *L. stoechas* aqueous extract responsible for nootropic activity.

## 2. Materials and Methods

### 2.1. Chemicals and Drugs

The following chemicals used were of analytical grade, and procured from Sigma Aldrich, MS Traders, Lahore Pakistan: 2,2-*diphenyl*-1-picrylhydrazyl (DPPH) (95%), 5,5′-dithiobis-*2*-nitrobenzoic acid (*DTNB*) (99%), 4,4-dithiodipyridine (98%), Folin–Ciocalteu reagent (FCR), trichloroacetic acid (TCA) (98%), thiobarbituric acid (TBA) (99%), tannic acid (99.9%), superoxide dismutase (SOD), sodium dodecyl sulfate, sodium carbonate (99%), rutin, reduced nicotinamide adenine dinucleotide (NADH), potassium dichromate (99.5%), potassium acetate (99%), phenyl methanesulfonate (PMS), nitro blue tetrazolium (NBT), *n*-butanol (99.8%), *n*-hexane (99%), methanol (99.8%), hydrogen peroxide (H_2_O_2_) (85%), gallic acid, ethanol, chloroform (99%), carboxy methyl cellulose (CMC), ascorbic acid (99%), aluminum chloride (99.9%), acetyl thiocholine iodide, silica gel-60, acetyl coenzyme-A, acetylcholine esterase, choline chloride (98%), EDTA, fast blue B salt, neostigmine sulfate, β-naphthyl acetate, and acetic acid. Piracetam was gifted from Jiangxi Yuehua Pharmaceutical, China, and scopolamine was obtained from Merck Pharmaceutical Pvt. Ltd. (Kenilworth, NJ, USA).

### 2.2. Extraction and Fractionation by Column Chromatography

*L. stoechas* L. (aerial parts) was purchased from a local market in Lahore, Pakistan. It was then identified by a botanist from the department of Botany, Government College University (GCU), Lahore. The specimen was preserved in the herbarium of the GCU and was assigned a voucher number: GC.Herb.Bot.3386. The extraction was conducted in methanol, by using simple maceration, as described in our previous study [15]. Then, fractional extraction was conducted by using different solvents, i.e. *n*-hexane, chloroform, ethyl acetate, and *n*-butanol. Finally, the aqueous layer left behind was obtained and the solvent was evaporated to obtain a semi-solid aqueous extract of *L. stoechas.* Column chromatography was used for the fractionation of the aqueous extract of *L. stoechas*. An appropriate-sized glass column was packed with slurry, which was made by dissolving almost 10 g of silica gel-60 in the same solvent as used for the mobile phase. Standard protocols were followed for the packing and running of the column. *L. stoechas* aqueous extract was dissolved in a small amount of solvent; that sample mixture was loaded in a column via a pipette. The space above the sample in the column was filled with solvent and the stopper was opened to obtain the separated fraction in a flask below the outlet of the column [16]. This way, different solvent systems were used for the separation of different constituents in the crude extract. Separated fractions were collected in the test tubes and labeled for further tests.

### 2.3. Anti-Cholinesterase Activity (In Vitro Assay)

In vitro anti-cholinesterase activity was performed on all fractions obtained through column chromatography by using an (NA-FB) microwell plate assay. The solution was prepared by dissolving β-naphthyl acetate (0.25 mg) in methanol (1 mL); 50 µL of this was mixed with 10 µL of plant extract. Then, 200 µL of acetylcholine esterase solution (3.33 U/mL) was poured in the reaction mixture by keeping the temperature of the mixture at 4 °C. It was incubated for 40 min at the same temperature and then 2.5 mg of fast blue B salt was dissolved in 1 mL of distilled water. Out of which, 10 µL was dropped into the incubated reaction mixture; a change in the solution color was observed. β-naphthyl acetate was used as a substrate, while fast blue B salt was used as a color reagent. The principle applies that β-naphthyl acetate is hydrolyzed into naphthol acetic acid by AChE. Naphthol then reacts with fast blue B salt, imparting a purple color to the mixture. No change in color of the solution indicates a strong anti-cholinesterase activity of the fraction, while a dark purple color indicates no inhibition of AChE [17].

### 2.4. GC–MS Analysis

The bioactive compounds present in the final fraction of *L. stoechas* were detected by performing gas chromatography mass spectroscopy (GC–MS) analysis by using GC–MS equipment (Agilent 6890N). The TR-5-MS capillary non-polar standard column, with dimensions of 30 Mts, an internal diameter of 0.25 mm, and film thickness of 0.25 µm, was used. Helium gas (99.99%) was used as carrier gas and mobile phase was run with a flow rate of 1 mL/min. The starting temperature of the oven was 40 °C, which was raised to 250 °C @ 10 °C/min. The sample was dissolved in methanol and an aliquot of 2 µL was injected by keeping the temperature of the injector and detector fixed at 250 °C and 280 °C, respectively, while the ion source temperature was fixed at 200 °C. Identification of the compound was by molecular mass and the structure of the compound by interpretation of the GC–MS standard library.

### 2.5. Behavioral and Biochemical Studies

Behavioral studies were performed by using elevated plus maze, light/dark test, and hole-board test models, using standard protocols. Biochemical studies were performed to assess the level of acetylcholinesterase (AChE), malondialdehyde (MDA), superoxide dismutase (SOD), catalase (CAT), and glutathione (GSH) in the brain homogenates of mice, of all groups, using standard protocols. Detailed procedures of these tests are described in a previously published paper in this series [15]. 

### 2.6. Animals

All studies were performed on male Swiss albino mice, which were provided standard living conditions, as described in a previous manuscript [15], while female albino mice (20–25 g) were used for toxicity studies.

### 2.7. Study Design for Behavioral/Biochemical Studies

For behavioral/biochemical studies, mice were divided into seven groups (*n* = 6) and were treated accordingly, as shown in Table 1. While ChAT was performed on five groups (*n* = 6) of mice, which were treated accordingly, as shown in Table 2.

### 2.8. Choline Acetyltransferase Activity (ChAT)

The reagent was prepared by mixing 10 µL of each sodium phosphate buffer (0.5 M, pH 7.2), sodium chloride (3 M), neostigmine sulfate (7.6 × 10^−4^ M), acetyl coenzyme-A (6.2 × 10^−3^ M prepared in 0.01N HCl), EDTA (1.1 × 10^−3^ M), choline chloride (1 M), and sodium chloride (3 M), and incubated at 37 °C for 5–10 min. Brain homogenate (100 µL) was mixed in the smallest volume of the reagent and the final volume was 0.2 mL. It was then boiled in a water bath for 2 min after incubating at 37 °C for 25 min. Then, oxygen-free distilled water was added to the mixture, which was then centrifuged at high speed to separate out the denatured protein. Then, 10 µL of 4,4-dithiodipyridine (10^−3^ M) was added to 0.5 mL of the supernatant and absorbance was read at 324 nm against the blank in a UV-visible spectrophotometer [18].

### 2.9. Acute Toxicity Studies

Acute toxicity study was performed on female albino mice according to OECD guidelines 423 2001. Initially, the pilot study was done on a small number of mice to find the dose range at which animals started to die. It was found that active fractions of *L. stoechas* (AfL.s) showed no death up to 300 mg/Kg/p.o. and showed 100% mortality at 500 mg/Kg/p.o. Thus, animals were divided into six groups (*n* = 5). Group I was normal control; animals in Groups II to VI were administered with AfL.s in wide, spaced doses, in ascending order (300, 350, 400, 450, and 500 mg/Kg/p.o., respectively). They were then observed for 24 h to find the number of mortalities. Finally, LD_50_ was calculated by using the following formula:LD_50_ = Least Lethal Dose − Σ (a × b)/n 

The animals who survived in different groups after administration of acute toxic dose were observed for two weeks for the assessment of physical and behavioral changes. Ataxia, blanching, convulsions, cyanosis, depression, hyperactivity, hypnosis, irritability, jumping, loss of traction, muscle spasm, piloerection, redness, rigidity, salivation, secretions, sedation, stimulation and straub reaction were the parameters that were observed after acute doses administration.

### 2.10. Statistical Analysis

The data are expressed as mean ± SEM. Student’s *t*-test analysis was applied on data with paired comparisons, and multiple comparisons were made by ANOVA followed by Dunnett’s test by using GraphPad Prism software version 7. Value of *p* < 0.05 was marked as significant.

## 3. Results

### 3.1. Fractionation by Column Chromatography

*L. stoechas* aqueous extract was fractionated further by column chromatography. In total, 55 fractions were obtained by using different solvent systems (based on polarity). The fractions were evaluated by thin layer chromatography (TLC), were combined, and again passed from the column; finally, there were 15 fractions (Figure 1).

#### Anti-Cholinesterase Activity (In Vitro)

In vitro testing indicated that fraction no. 6 showed the best enzyme inhibition among all the fractions, as expressed in Table 3. Fraction no. 6 was named an active fraction of *L. stoechas* (AfL.s) and was tested for chemical analysis by GC–MS. Furthermore, in-vivo studies (behavioral and biochemical) were performed on mice to find the memory enhancing effects of AfL.s.

### 3.2. GC–MS Analysis of AfL.s

GC–MS analysis indicated that AfL.s contained the phenethylamine group of compounds and cholestan-7-one. The details are shown in Table 4 and spectrum is given in Appendix A.

### 3.3. Behavioral Studies (Effect of AfL.s on EPM, Light/Dark Test, and Hole-Board Paradigm in Mice)

Results of behavioral studies indicated that animals treated with AfL.s significantly (*p* < 0.001) reduced the initial transfer latencies (ITL) and retention transfer latencies (RTL) in comparison to the amnesic control group (Figure 2). Similarly, inflexion ratio (IR) calculated from ITL and RTL values, indicated that active fraction-treated mice showed maximum IR value (0.17 ± 0.04) in comparison to scopolamine-treated mice (−0.19 ± 0.04). Thus, higher IR values indicated significant (*p* < 0.001) improvement of memory in AfL.s-treated mice.

Results of the light/dark test indicated that AfL.s-treated mice spent most of the time in the dark portion of the apparatus, on the first and second day of observation, in comparison to the amnesic control group. This finding is based on the principle that animals in the amnesic control group forgot to find the dark area of the apparatus while the standard control and AfL.s-treated mice retained their memory of exploration. Thus, it is clear that AfL.s significantly (*p* < 0.001) improved the memory in AfL.s-treated mice (Figure 2).

Similarly, results of hole-board paradigm indicated that amnesic control animals significantly reduced the number of hole-pokings while standard control and AfL.s-treated mice retained their memory of exploration and showed a significantly (*p* < 0.001) increased number of hole-pokings on both the first and second day of observation (Figure 2).

### 3.4. Biochemical Studies (Effect of AfL.s on Levels of AChE, MDA, SOD, CAT, and GSH in Mice Brains)

Biochemical studies indicated that the level of acetylcholinesterase (AChE) was significantly (*p* < 0.001) reduced in group-II (scopolamine treated) animals, while group-VII animals (AfL.s 18 mg/Kg/p.o.) showed maximum inhibition of AChE among all the groups (Figure 3A). Similarly, the level of MDA is reduced significantly (*p* < 0.001) in AfL.s-treated mice as compared to the amnesic control group (which showed the highest MDA level) (Figure 3B). It was observed that the levels of SOD, CAT, and GSH increased significantly in group-V (AfL.s 9 mg/Kg/p.o.) among all groups (Figure 3C–E). This observation leads to the fact that AfL.s possesses strong antioxidant activity when used in low doses.

### 3.5. Effect of AfL.s on ChAT Activity

The level of ChAT was observed, 11.85 ± 0.92, 7.59 ± 0.76, 18.13 ± 1.23, 16.70 ± 1.16, 12.10 ± 1.45, 11.63 ± 0.66, 10.84 ± 1.22, and 7.08 ± 0.68 μmol/min/mg, from group-I to V, respectively. Thus it is clear that animals treated only with AfL.s (18 mg/Kg/p.o.) for seven consecutive days showed the best elevation in ChAT levels as compared to the normal control group, with the level of significance as (*p* < 0.01), as shown in Figure 4.

### 3.6. Acute Toxicity Study

Acute toxicity study performed on six groups (*n* = 5) of female albino mice indicated that AfL.s produced no death up to a dose of 350 mg/Kg/p.o. Animals began to die when the dose increased above 350 mg/Kg/p.o. All the animals died when they were administered with a single oral dose 500 mg/Kg of AfL.s. All the details and calculations for median lethal dose LD_50_ are provided in (Table 5). LD_50_ for AfL.s was calculated as 325 mg/Kg/p.o. The animals who survived after the administration of an acute single dose of AfL.s 450 mg/Kg/p.o. were kept in observations for two weeks for the assessment of behavioral and physiological changes. It was observed that animals positively exhibited stimulation, straub reaction. salivation, rigidity, piloerection, other secretions, muscle spasm, loss of traction, jumping, irritability, hyperactivity, convulsions, blanching, and ataxia, while, sedation, redness, ptosis, hypnosis, depression, and cyanosis were not observed in the mice (Table 6).

## 4. Discussion

Aromatic plants of the Mediterranean region have advanced medicinal values. Alcoholic and hydro-alcoholic extracts of many aromatic plants of Asteraceae, Apiaceae, and Lamiaceae families have been used extensively for their wide variety of pharmacological activities. Genus Lavandula contains as much as 39 species, including two economically well renowned species—*L. stoechas* and *L. angustifolia*. Hydroalcoholic extracts of *L. stoechas* contain catechic tannins, flavonoids, coumarins, sterols, mucilages, and leucoanthocyanins [19]. Different compounds, such as α-amyrin, β-sitosterol, α-amyrin acetate, oleanolic acid, vergatic acid, ursolic acid, erythrodiol, lupeol, luteolin, vitexin, acacetin, lavanol, 7-methoxy coumarin, and two longipinane derivatives (longipin-2-ene-7β,9α-diol-1-one-9-monoacetate and longipin-2-ene-7β,9α-diol-1-one) have been isolated from areal parts of *L. stoechas* [20]. *L. stoechas* exhibits a wide array of pharmacological activities; research shows that it boosts memory in albino mice [15]. This is an original research work specifically designed to report the biomolecules of *L. stoechas,* which boosts memory in mice brains. In this study, the *L. stoechas* aqueous extract was purified, and chemical characterization of AfL.s by GC–MS indicated the presence of two main compounds (phenethylamine and α-tocopherol) (Appendix A, Table 4). Past studies have scientifically proven the anticholinesterase activity of phenethylamine [21] and the antioxidant potential of cholestan-7-one, usually named α-tocopherol [22]. 

Phenethylamine has been scientifically proven as a brain neuromodulator and a strong inhibitor of AChE [23]. Phenethylamine is composed of an aromatic ring to which a side chain of two carbons having amine at the terminal position is attached. Substitution of alkyl groups at different positions on the phenyl ring would attribute to its strong neuromodulative and psychoactive activities. Thus, overall cognition and brain performance is enhanced by phenethylamine [21,24].

The results of the behavioral studies concluded that AfL.s significantly (*p* < 0.001) enhanced the retention power and learning capacity of the mice brains. Similarly, treatment of animal with AfL.s showed significant (*p* < 0.001) reduction in the level of AChE (Figure 3A). Inhibiting AChE improves cholinergic transmission and relieves the patient of memory loss [25].

On the other hand, the level of choline acetyltransferase (ChAT) was elevated (Figure 4) in mice brains. Both of these findings strongly suggest that the level of acetylcholine (ACh) increased in mice brains by dual mechanisms. This effect would be due to the action of phenethylamine on trace amine-associated receptors (TAARs), which are abundantly present in the brain, pituitary glands, kidney, liver, and stomach [26]. It has been proposed that phenethylamine binds with G-protein (either G_s_ or G_q_ subunit) coupled receptors (TAARs) and enhances memory and cognition [27,28].

The possible antioxidant mechanism of AfL.s is due to the presence of α-tocopherol, which not only promotes the glutathione level in the brain, but also causes attenuation of reactive oxygen species [22]. Furthermore, α-tocopherol is responsible for the formation of the stable and inert tocopheroxyl radical, by reacting with lipid peroxyl radical (LOO˙), as shown in Equation [29].
LOO˙ + α-tocopherol–OH → LOOH + α-tocopherol–O˙

Loss of memory may take place, either due to reduction in the level of acetylcholine [30] or by deposition of the β-amyloid protein [31] in the cerebral cortex and hippocampus of the brain [32]. Moreover, severe oxidative damage to neuronal circuits in the brain is another leading cause of memory loss [33,34], which is exhibited in scopolamine-treated mice [35]. The findings of the current study indicate that scopolamine-treated mice showed marked elevation of AChE (Figure 3A) and reduction in ChAT (Figure 4). Oxidative stress induced by scopolamine is the main factor behind this enzyme disturbance [36]. The treatment of mice with AfL.s not only reduced the level of AChE (Figure 3A), but also significantly (*p* < 0.001) boosted the level of ChAT (Figure 4) in mice brains. It is proposed that elevation in ChAT levels by AfL.s is caused by the antioxidant action of α-tocopherol on the brain. Similarly, phenethylamine present in AfL.s is responsible for the enzyme-mediated release of ACh in the brain [37]. The results also clearly indicate that animals only treated with AfL.s, without prior or subsequent administration of scopolamine, produced the highest elevation of ChAT levels in the brain (Figure 4). 

Acute toxicity study indicated that AfL.s is toxic when used in high doses (≥400 mg/Kg/p.o.), which produces hyperactivity, hyperstimulation, ataxia, seizures, and ultimate death (Table 6). This toxicity is due to the primary toxic effect of high doses of phenethylamine, which is responsible for headaches, confusions, hallucination, seizures, and ultimately death in human beings [38]. Restlessness, diarrhea, headache, aggression, and tremors are mild side effects, which may be observed with the overdoses of phenethylamine [39]. 

Past studies have reported that high toxic doses (125–200 mg/Kg/i.p.) of phenethylamine produced very severe seizures and ultimate deaths due to cardiac arrest and overstimulation of the brain [40]. The LD_50_ for AfL.s was calculated as 325 mg/Kg/p.o. (Table 5), which indicated that it had a broad therapeutic index.

## 5. Conclusions

Two principle compounds—α-tocopherol and phenethylamine, present in *L. stoechas*—are responsible for the attenuation of dementia. α-tocopherol reduces oxidative stress of the free radicals in mice brains while phenethylamine enhances the level of acetylcholine in the hippocampus of mice brains. Thus, it is concluded that *L. stoechas* L. can be used as a memory enhancer. Further studies are needed to elaborate on its detailed mechanisms, in regards to enhancement of memory and toxicity profile.

## Figures and Tables

**Figure 1 plants-10-01259-f001:**
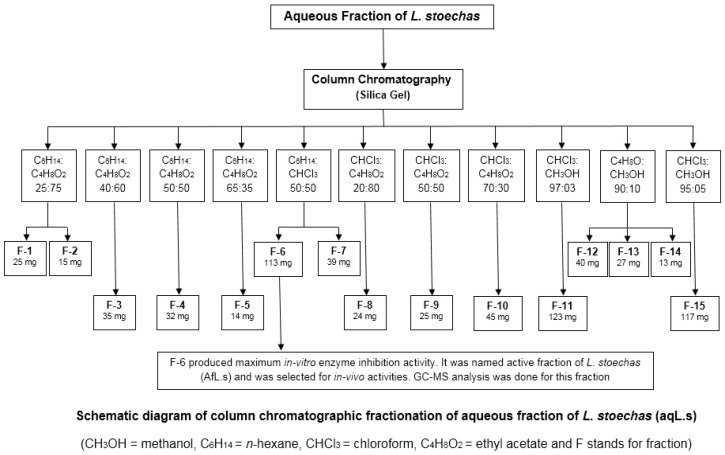
General scheme of fractionation by column chromatography.

**Figure 2 plants-10-01259-f002:**
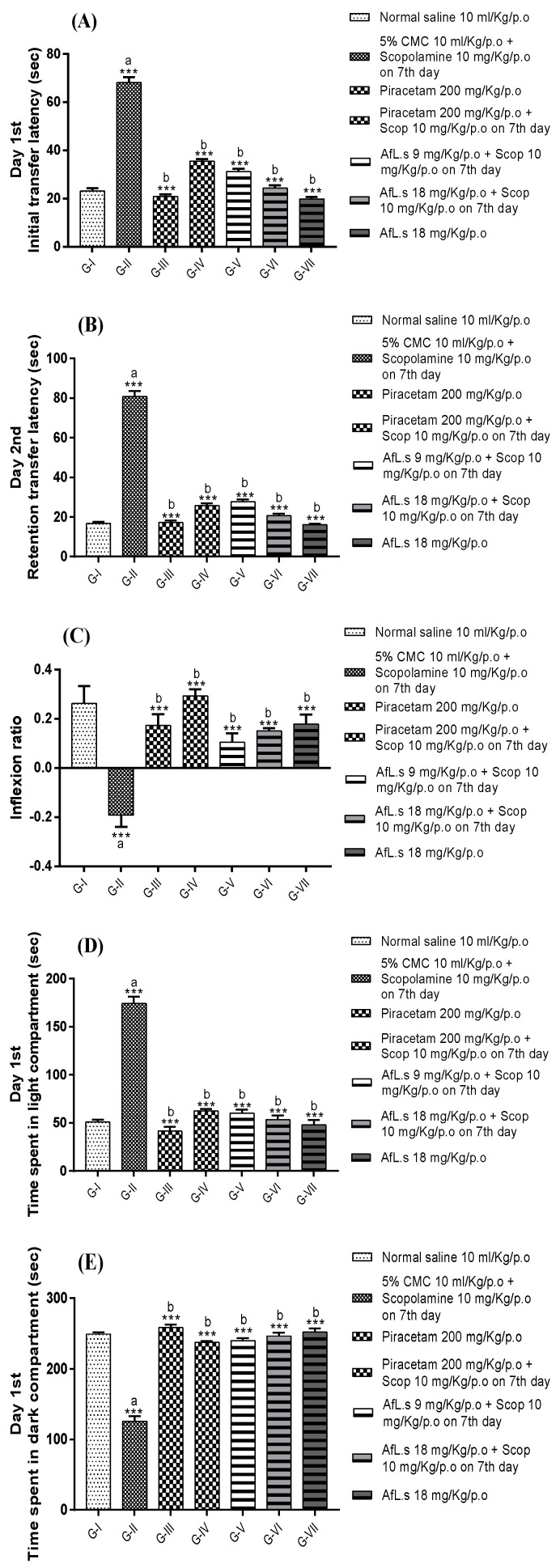
Effect of AfL.s on (**A**) initial transfer latency; (**B**) retention transfer latency; (**C**) inflexion ratio in the elevated plus maze paradigm; (**D**) time spent (sec) in the light compartment on day 1; (**E**) time spent (sec) in the dark compartment on day 1; (**F**) time spent (sec) in the light compartment on day 2; (**G**) time spent (sec) in the dark compartment on day 2; (**H**) number of hole pokings by mice on day 1, and (**I**) number of hole pokings by mice on day 2. Data are presented as mean ± SEM (*n* = 6) and one-way ANOVA (Dunnett’s test) was applied by comparing G-II to G-I (presented by “a” on the bar). All other groups were compared to G-II (presented by “b” on the bar). The signs *ns,* *, ** and *** presented the *p* values as ≥0.05, ≤0.05, ≤0.01, and ≤0.001, respectively).

**Figure 3 plants-10-01259-f003:**
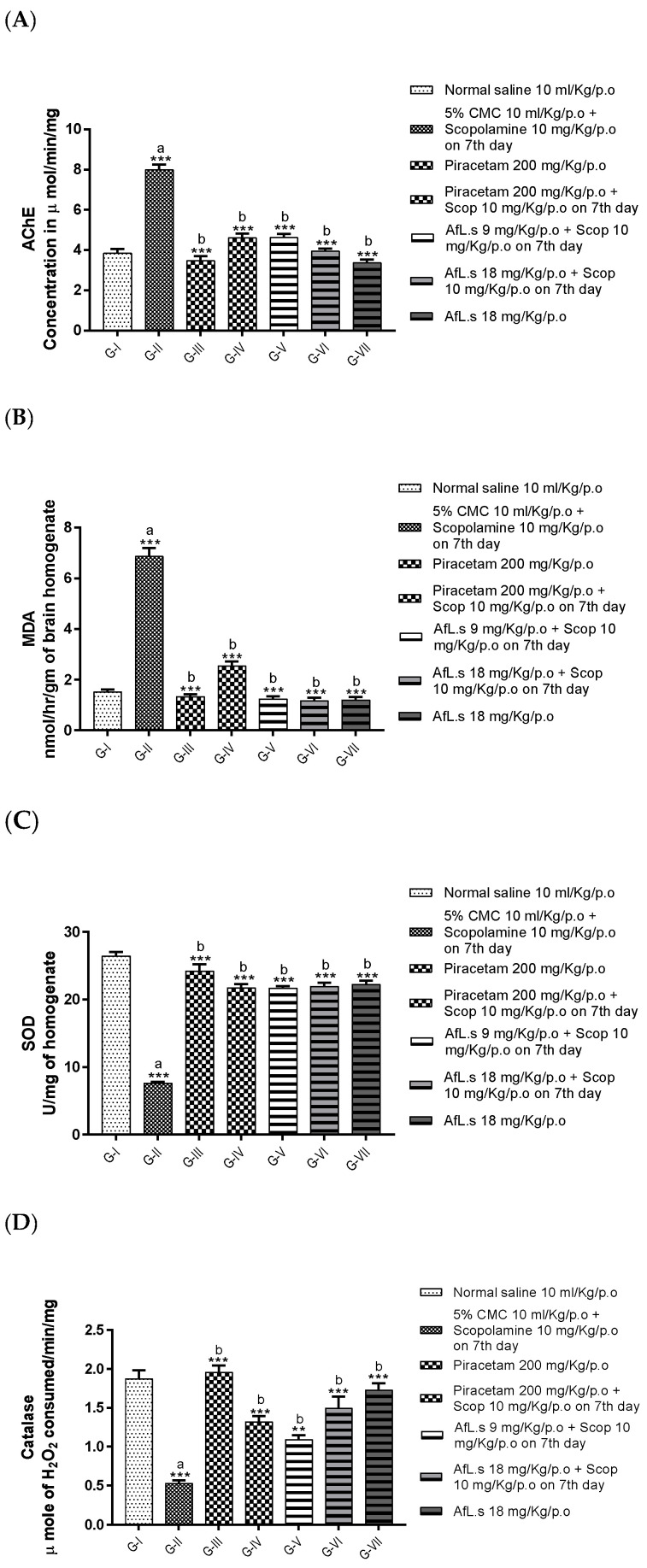
Effect of AfL.s on concentration of (**A**) acetylcholinesterase (AChE); (**B**) MDA; (**C**) SOD; (**D**) CAT; and (**E**) GSH in brain homogenate. Data are presented as mean ± SEM (*n* = 6) and one-way ANOVA (Dunnett’s test) was applied by comparing G-II to G-I (presented by “a” on bar). All other groups were compared to G-II (presented by “b” on bar). The signs *ns*, *, ** and *** presented the *p* values as ≥0.05, ≤0.05, ≤0.01, and ≤0.001, respectively).

**Figure 4 plants-10-01259-f004:**
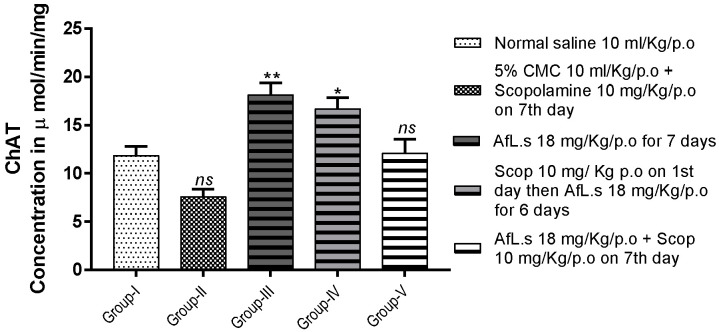
Effect of AfL.s on concentration of ChAT in mice brains. Data are presented as mean ± SEM (*n* = 6) and one-way ANOVA (Dunnett’s test) was applied by comparing all groups with G-I. The signs *ns*, *, ** and *** presented the *p* values as ≥ 0.05, ≤ 0.05, ≤ 0.01, and ≤ 0.001 respectively.

**Table 1 plants-10-01259-t001:** Study design for behavioral and biochemical studies.

Groups	Treatment from Days 1–7	Treatment on Day 7, 45 min after Administration of the Last Dose
G-I (Normal Control)	Normal saline 10 mL/Kg/p.o.	- - - - - -
G-II (Amnesic Control)	5% CMC 10 mL/Kg/p.o.	Scopolamine (10 mg/Kg/p.o.)
G-III (Standard Control-A)	Piracetam 200 mg/Kg/p.o.	- - - - - -
G-IV (Standard Control-B)	Piracetam 200 mg/Kg/p.o.	Scopolamine (10 mg/Kg/p.o.)
G-V (Experimental Control-I)	AfL.s 9 mg/Kg/p.o.	Scopolamine (10 mg/Kg/p.o.)
G-VI (Experimental Control-II)	AfL.s 18 mg/Kg/p.o.	Scopolamine (10 mg/Kg/p.o.)
G-VII (Experimental Control-III)	AfL.s 18 mg/Kg/p.o.	- - - - - -

Doses were prepared by suspending AfL.s in CMC (5%) and by dissolving piracetam and scopolamine in normal saline. Then, behavioral studies were performed on days 7 and 8, and animals were sacrificed for the performance of biochemical studies on the eigth day after completing behavioral trials.

**Table 2 plants-10-01259-t002:** Study design for the assessment of choline acetyltransferase (ChAT) activity.

Groups	Treatment		
	Day 1	Days 2–6	Day 7
G-I	Normal Saline (10 mL/Kg/p.o.)	Normal Saline (10 mL/Kg/p.o.)	Normal Saline (10 mL/Kg/p.o.)
G-II	5% CMC (10 mL/Kg/p.o.)	5% CMC (10 mL/Kg/p.o.)	Scopolamine (10 mg/Kg/P.O)
G-III	AfL.s (18 mg/Kg/p.o.)	AfL.s (18 mg/Kg/p.o.)	AfL.s (18 mg/Kg/p.o.)
G-IV	Scopolamine (10 mg/Kg/p.o.)	AfL.s (18 mg/Kg/p.o.)	AfL.s (18 mg/Kg/p.o.)
G-V	AfL.s (18 mg/Kg/p.o.)	AfL.s (18 mg/Kg/p.o.)	AfL.s (18 mg/Kg/p.o.) + Scopolamine (10 mg/Kg/p.o.)

Two hours after administration of the last dose, the animals were sacrificed by using chloroform to get their brain. Then brain homogenates were formed according to standard procedure [15] and ChAT activity was determined by the above-described spectroscopic method.

**Table 3 plants-10-01259-t003:** Anti-cholinesterase activity (in vitro) shown by different fractions of *L. stoechas*.

No.	Fractions	Color of Solution	Inhibition of AChE
1	F-1	Dark purple	No
2	F-2	Dark purple	No
3	F-3	Dark purple	No
4	F-4	Dark purple	No
5	F-5	Dark purple	No
6	F-6	No color change	Very strong
7	F-7	Light purple	Mild
8	F-8	Dark purple	No
9	F-9	Dark purple	No
10	F-10	Dark purple	No
11	F-11	Dark purple	No
12	F-12	Light purple	Mild
13	F-13	Light purple	Mild
14	F-14	Light purple	Mild
15	F-15	Dark purple	No

Fraction no. 6 (F-6) possessed strong anti-cholinesterase activity, so it was selected for further chemical and in-vivo studies.

**Table 4 plants-10-01259-t004:** Compounds detected in AfL.s by GC–MS.

	Compound Name	Molecular Formula	Molecular Weight (g/mol)	Mass Peak	Retention Time (min)
1	Phenethylamine, N methyl-beta 3,4 (trimethylsiloxy)	C_18_H_37_NO_3_Si_3_	399	50	21.167
2	Cholestan-7-one	C_29_H_50_O_2_	430	56	24.083
3	Phenethylamine	C_18_H_37_NO_3_Si_3_	399	50	21.167
4	N-Methyladrenaline	C_19_H_39_NO_3_Si_3_	413	50	21.167
5	Benzeneacetic acid	C_20_H_42_O_5_Si_4_	472	50	21.167

**Table 5 plants-10-01259-t005:** Calculation of median lethal dose LD_50_ of AfL.s.

Groups	Dose Difference (a)	Mortality	Mean Mortality (b)	(a × b)
G-I (Normal Control)	0	0	0	0
G-II (AfL.s 300 mg/Kg/p.o.)	300	0	0	0
G-III (AfL.s 350 mg/Kg/p.o.)	50	0	0	0
G-IV (AfL.s 400 mg/Kg/p.o.)	50	2	2 + 0/2 = 1	50
G-V (AfL.s 450 mg/Kg/p.o.)	50	3	3 + 2/2 = 2.5	125
G-VI (AfL.s 500 mg/Kg/p.o.)	50	5	5 + 3/2 = 4	200
				**Σ (a × b) = 375**

Σ(a × b)/*n* = 375/5; LD_50_ = Least lethal Dose − Σ(a × b)/*n*; 400 − 375/5 = 325 mg/Kg.

**Table 6 plants-10-01259-t006:** Effect of acute toxic dose of AfL.s 450 mg/Kg/p.o. on the behavior and physiology of mice.

Behavioral Changes	Days													
	1	2	3	4	5	6	7	8	9	10	11	12	13	14
Straub Reaction	-	-	+	+	+	+	+	+	+	-	-	-	-	-
Stimulation	+	+	+	+	+	+	+	+	+	+	-	-	-	-
Sedation	-	-	-	-	-	-	-	-	-	-	-	-	-	-
Secretions	+	+	+	+	+	+	+	-	-	-	-	-	-	-
Salivation	+	+	+	+	+	-	-	-	-	-	-	-	-	-
Rigidity	+	+	+	+	+	+	+	-	-	-	-	-	-	-
Redness	-	-	-	-	-	-	-	-	-	-	-	-	-	-
Ptosis	-	-	-	-	-	-	-	-	-	-	-	-	-	-
Piloerection	+	-	-	-	-	-	-	-	-	-	-	-	-	-
Muscle Spasm	+	+	+	+	-	-	-	-	-	-	-	-	-	-
Loss of Traction	-	-	-	-	-	-	+	+	+	+	+	+	+	+
Jumping	+	+	+	+	+	+	+	+	+	+	+	-	-	-
Irritability	+	+	+	+	+	+	+	+	+	+	+	+	+	+
Hypnosis	-	-	-	-	-	-	-	-	-	-	-	-	-	-
Hyperactivity	+	+	+	+	+	+	+	+	+	+	+	+	+	+
Depression	-	-	-	-	-	-	-	-	-	-	-	-	-	-
Cyanosis	-	-	-	-	-	-	-	-	-	-	-	-	-	-
Convulsions	+	+	+	-	-	-	-	-	-	-	-	-	-	-
Blanching	+	+	+	+	+	+	-	-	-	-	-	-	-	-
Ataxia	+	+	+	+	+	+	+	+	-	-	-	-	-	-

“-“Sign indicates absence of effect while “+” sign indicates the presence of effect.

## Data Availability

All of the data are available and can be produced upon request.

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
