# Peer review of "Biomolecular Evaluation of Lavandula stoechas L. for Nootropic Activity"

_plants, 2021, doi:10.3390/plants10061259_

Round 1

Reviewer 1 Report

This publication is of interest and brings additional and substantial scientific results from work recently published by authors in Frontiers in pharmacology (A Mushtaq et al. Lavandula stoechas (L) a very potent antioxidant attenuates dementia in scopolamine induced memory deficit mice, 2018, 9, 1375.).

The authors treat the advances of the biomolecules diversity present in L. stoechas L. responsible for its memory enhancing potential, by fractioning the aqueous extract of L. stoechas into fifteen fractions and by testing them for in-vitro anti-cholinesterase activity to find the active fraction of L. stoechas. Authors used a wide range of experimental and theoretical techniques and conclude that α-tocopherol and phenethylamine present in L. stoechas enhance the memory in animal models by reducing oxidative stress and by enhancing the level of acetylcholine in hippocampus of the mice brain. Overall, the manuscript is rich and interesting; and the paper structure is well-knit and suitable for publication in the journal, after minor revisions. The comments are listed as the following points:

  1. In the abstract and all manuscript: An aromatic amine is an organic compound consisting of an aromatic ring directly attached to an amine, it means that Phenethylamine cannot be considered as an aromatic amine but it is better to use the word primary amine.
  2. P 1: Line 4: There is a status that should not be there “Rector Research”. Please check.
  3. P 1 : Line 35-36: “madicinal” should be “medicinal”, and also,“Reagon” should be “Region”. Please check
  4. P 2: Reference 3 is not accessible
  5. P 2: Line 43: “varirty” should be “variety”
  6. Figure 2 : The resolution of figure 2 should be improved
  7. Please, improve the resolution of figure 6.
  8. P15: Line 347: “phenethylamene” should be “phenethylamine”
  9. It will be very important to mention two recent references on the Phytopharmacological and Phytochemicals of Lavandula stoechas L: Ref. 1: Bousta, D., & Farah, A. (2020). A Phytopharmacological review of a Mediterranean plant: Lavandula stoechas L. Clinical Phytoscience6(1), 9.                                                                              Ref. 2: Héral, B., Stierlin, É., Fernandez, X., & Michel, T. (2020). Phytochemicals from the genus Lavandula: a review. Phytochemistry Reviews, 1-21.

Author Response

The Editor

Plants

Subject: Submission of revised manuscript No. Plants: 1224126

Dear Sir

It is stated that I want to submit revised article entitled, “Memory enhancing potential of Lavandula stoechas L.: Evidence-based approach” for publication in your esteemed Journal. We are highly thankful to referees whose comments helped in improving this manuscript. We have revised the entire manuscript for language as well as for proper flow of the information. Mostly these are the reviewers’ observations which are addressed in the point by point rebuttal file and also incorporated the same in the text.

Below is response to referee comments:

Reviewer 1

Comment: This publication is of interest and brings additional and substantial scientific results from work recently published by authors in Frontiers in pharmacology (A Mushtaq et al. Lavandula stoechas (L) a very potent antioxidant attenuates dementia in scopolamine induced memory deficit mice, 2018, 9, 1375.).Overall, the manuscript is rich and interesting; and the paper structure is well-knit and suitable for publication in the journal, after minor revisions.

Authors response: Thank you for encouraging comments.

Comment: An aromatic amine is an organic compound consisting of an aromatic ring directly attached to an amine, it means that Phenethylamine cannot be considered as an aromatic amine but it is better to use the word primary amine.

Authors response: Corrected

Comment: P 1Line 4: There is a status that should not be there “Rector Research”. Please check.

Authors response: Corrected

Comment: P 1 : Line 35-36: “madicinal” should be “medicinal”, and also,“Reagon” should be “Region”. Please check

Authors response: Corrected

Comment: P 2: Reference 3 is not accessible

Authors response: Accessible. Kindly check this link. PMID: 10904159 DOI: 10.1016/s0378-8741(99)00198-1

Comment: P 2: Line 43: “varirty” should be “variety”

Authors response: Corrected

Comment: Figure 2 : The resolution of figure 2 should be improved

Authors response: Figure has maximum resolution as it was captured from the hard print of chromatogram.

Comment: Please, improve the resolution of figure 6.

Authors response: figure 6 has been removed from the manuscript as per recommendation of reviewer no 02.

Comment: P15Line 347: “phenethylamene” should be “phenethylamine”

Authors response: Corrected.

Comment: It will be very important to mention two recent references on the Phytopharmacological and Phytochemicals of Lavandula stoechas L: Ref. 1: Bousta, D., & Farah, A. (2020). A Phytopharmacological review of a Mediterranean plant: Lavandula stoechas L. Clinical Phytoscience6(1), 9.                                                                              Ref. 2: Héral, B., Stierlin, É., Fernandez, X., & Michel, T. (2020). Phytochemicals from the genus Lavandula: a review. Phytochemistry Reviews, 1-21.

Authors response: Inserted.

Reviewer 2

Comment: In the original article entitled “Memory enhancing potential of Lavandula stoechas L.: Evidence-based approach” by dr  Mushtaq et al., the Authors investigated the effects of biomolecules extracted from Lavandula stoechas  using both biochemical tests in vitro as well as behavioural studies in mice model, focusing on memory enhancing potential of the extract. In my opinion, the paper has reported interesting results, however I have one major and several minor comments.

Authors response: Thank you for encouraging comments.

Comment: Discussion should focus on explaining and commenting the results but not on presenting the results once again. However, the Authors frequently address the specific figures, the tables and provide statistical significance. Please see: Lines 309-314; lines 317-326; lines 330-331; line 345, lines 362-370 etc.In my opinion, the discussion should be rewritten.

Authors response: Discussion has been modified.

Comment: Line 4: Is “Rector Research” the name of the author?

Authors response: Corrected

Comment: Line 35: “medicinal” -> “medicinal”

Authors response: Corrected

Comment: Line 43: “varirty” -> “variety”

Authors response: Corrected

Comment: Lines “in-vitro” -> “in vitro”

Authors response: Corrected

Comment: Fig 2. The graphic quality and resolution of the Fig 2. is very low. It should be improved.

Authors response: Figure has maximum resolution as it was captured from the hard print of chromatogram.

Comment: Fig. 6. The chemical structure of phenethylamine is well known. I do not see any reason to present it in the discussion as the Fig. 6. I advice to remove the Fig 6. from the article.

Authors response: Removed

Reviewer 3

Abstract

Comment: " Evidence-based approach" is not suitable for the title

Authors response: Other reviewer has not raised objection on title. Kindly suggest some appropriate word

Comment: The graphical abstract is highly recommended.

Authors response: Graphical Abstract has been provided

Comment: The authors would add the originality of their work.

Authors response: This is the original research work.

Comment: Could you abbreviate the species of the plant throughout the manuscript?

Authors response: Corrected

Comment: GC-MS is used for chemically analysis of essential oils not the aqueous extract

Authors response: Yes GC-MS is used mostly for analysis of essential oils. In this study we fractionated the aqueous extract of L. stoechas by using different combinations of polar and non-polar solvents. Thus the fraction which best inhibited AChE Was selected and then GC-MS was used for identification of compound. Since the fraction was purified so it was analyzed by GC-MS by using methanol as solvent system.

Comment: The abstract is confusing, please re-write?

Authors response: In our opinion it is written in easy language and in a sequence. It may be acceptable in as such form.

Comment: The authors could use more specific keywords.

Authors response: Corrected

Introduction

Comment: Please, check the structure and the grammar of the first sentence.

Authors response: Corrected

Comment: Lines 35-36, please correct “madicinal” and “reagon”.

Authors response: Corrected

Comment: Kindly, focus on α-tocopherol and phenethylamine isolation, identification, and the different pharmacological effects.

Authors response: Not found any suitable methodology.

Materials and methods

Comment: Could you explain briefly the extraction procedure.

Authors response:  The extraction procedure is written in brief in this article. The detailed extraction procedure is explained in detail in previous published Article, the reference of which is given in methodology section as reference no 15. (The extraction was done in methanol by using simple maceration as described in our previous study [15])

Comment: Ethyl acetate is slightly less polar than chloroform. Why did you use both of them?

Authors response: I performed sequential extraction. And to perform this I observed all the standard procedures which are mentioned in literature. In literature, most of the time the same sequence of solvents is used for sequential extraction.

Comment: How did you evaporate the solvent?

Authors response:  Solvent was evaporated in rotary evaporated under reduced pressure (explained in previous study ref no. [15].

Comment: Which solvent was used to dissolve the extract and fill the column space?

Authors response:  Different solvents were used in combinations in different ratios as mentioned in Figure 1 in manuscript.

Comment: What are the parameters you used when collecting the separated fractions?

Authors response:  As mentioned in reference 16

Comment: Line 89, please correct “essay”.

Authors response:  Corrected

Comment: For each assay, clarify if it was conducted on the extract or the fraction.

Authors response:  It is mentioned in the heading 2.2 that separated fractions were collected in the test tubes and labeled for the further tests. Mean that assay was done on fraction.

Comment: Is it rational to analyze the aqueous fraction using GC-MS?

Authors response:  Aqueous solvent is inappropriate to be used in GC-MS. But in our study we purified the aqueous extract by column chromatography, then we analyzed that purified fraction in GC-MS by using methanol as solvent system.

Comment: Please, make sure that you cite an appropriate reference in each experiment.

Authors response: All references are correctly mentioned.

Comment: “using elevated plus maze, light/dark test and hole board test”, please explain in more details.

Authors response:  They are well explained in our previous study [15]

Comment: Please explain briefly the biological assays of the manuscript.

Authors response:  Behavioral studies were performed by using elevated plus maze, light/dark test and hole board test models by using standard protocols. While biochemical studies were performed to assess the level of acetylcholinesterase (AChE), Malondialdehyde (MDA), Superoxide Dismutase (SOD), Catalase (CAT) and Glutathione (GSH) in brain homogenates of mice of all groups by using standard protocols. Detailed procedures of all of these tests are described in previous published paper of this series [15]. To avoid repetition and plagiarism the details are not mentioned in this manuscript. The reference is inserted for the readers to access the details.

Results and discussion

Comment: Better move figure 1 to the materials and methods part.

Authors response: Moved to Methodology

Comment: How is the first solvent ratio is 25:50 in figure 1?

Authors response: The ratio is 25:75. There is typographic mistake.

Comment: Could you double-check the three last solvent systems?

Authors response: I have checked, it is all ok.

Comment: Could you add the charts of the identified metabolites made by GC-MS?

Authors response: The compounds are shown in table 4.

Comment: Is silane one of the plant metabolites?

Authors response:  No this is not. It has been removed.

Comment: what about the bioactivity of the other identified metabolites?

Authors response: Our focus was to prove the memory enhancing potential of L. stoechas. We found that the phenethylamine and alpha tocophorel have historical evidence of enhancement of memory.

Comment: Expand on the conclusion part and add the future perspectives.

Authors response:  Added

Reviewer 4

Comments: The present paper “Memory enhancing potential of Lavandula stoechas L.: Evidence-based approach”is very interesting and innovative;moreover, it is very smooth and pleasant to read.

Authors response: Thank you for encouraging comments.

Comments: Do you think that plants belonging to species other than Lavandula stoechas may have similar biological properties?

Authors response: Biological properties of the plants are dependent on constituents. There may be the chance that the resembling constituents are found in the spices of the same genus. So, there may be the resemblance of biological activities due to the presence of similar constituents in the plant.

Comments: Have you measured the antioxidant properties and polyphenolic content of the extract in vitro? If so, please indicate the results in the work.

Authors response: The detailed extraction procedure including in vitro antioxidant studies have been expressed in previously published paper (Front. Pharmacol., 23 November 2018 | https://doi.org/10.3389/fphar.2018.01375) of this series. In this manuscript the main constituents of L. stoechas, responsible for enhancement of memory is reported.

Comments: Reagents not used in the work are indicated in Chemicals and Drugs.

Authors response: Numbers of reagents were used and it was not possible to mention the details of the reagents in the manuscript.

Thank you once again for your valuable comments. I am available if there are any further queries.

--

Best regards,

Reviewer 2 Report

In the original article entitled “Memory enhancing potential of Lavandula stoechas L.: Evidence-based approach” by dr  Mushtaq et al., the Authors investigated the effects of biomolecules extracted from Lavandula stoechas  using both biochemical tests in vitro as well as behavioural studies in mice model, focusing on memory enhancing potential of the extract. In my opinion, the paper has reported interesting results, however I have one major and several minor comments.

Major comment:

Discussion should focus on explaining and commenting the results but not on presenting the results once again. However, the Authors frequently address the specific figures, the tables and provide statistical significance. Please see: Lines 309-314; lines 317-326; lines 330-331; line 345, lines 362-370 etc.

In my opinion, the discussion should be rewritten.

Minor comments:

Line 4: Is “Rector Research” the name of the author?

Line 35: “medicinal” -> “medicinal”

Line 43: “varirty” -> “variety”

Lines “in-vitro” -> “in vitro”

Fig 2. The graphic quality and resolution of the Fig 2. is very low. It should be improved.

Fig. 6. The chemical structure of phenethylamine is well known. I do not see any reason to present it in the discussion as the Fig. 6. I advice to remove the Fig 6. from the article.

Author Response

(The authors gave the same response as above.)

Reviewer 3 Report

May 17, 2021

Journal: Plants

Title: Memory enhancing potential of Lavandula stoechas L.: Evidence-based approach

Authors: Aamir Mushtaq, Rukhsana Anwar, Umar Farooq Gohar, Mobasher Ahmad, Marc Romina Alina, Rector Research, Marius Irimie, Elena Bobescu

Dear Editor:

The authors have investigated the potential of L. stoechas biomolecules for the management of dementia, in-vitro anti-cholinesterase activity, as well as in-vivo (behavioral and biochemical) studies. The manuscript could be accepted after major revision.

Hesham El-Seedi, professor,
Associate Editor

Journal of Advanced Research, I.F.= 6.99

--------------------------------------

Department of Molecular Biosciences,

The Wenner-Gren Institute,

Stockholm University, S-106 91,

Stockholm, Sweden

Tel: +46-700 43 43 43

E-mail address: hesham.elseedi@su.se

-----------------------------------------

Pharmacognosy Division,

BMC, Uppsala University,

SE-751 23 Uppsala, Sweden

Email: hesham.el-seedi@farmbio.uu.se

Comments to authors:

Abstract:

  1. " Evidence-based approach" is not suitable for the title
  2. The graphical abstract is highly recommended.
  3. The authors would add the originality of their work.
  4. Could you abbreviate the species of the plant throughout the manuscript?
  5. GC-MS is used for chemically analysis of essential oils not the aqueous extract
  6. The abstract is confusing, please re-write?
  7. The authors could use more specific keywords.

Introduction:

  1. Please, check the structure and the grammar of the first sentence.
  2. Lines 35-36, please correct “madicinal” and “reagon”.
  3. Kindly, focus on α-tocopherol and phenethylamine isolation, identification, and the different pharmacological effects.
  4. The authors could get benefit from this reference:

Arya, A., Chahal, R., Rao, R., Rahman, M. H., Kaushik, D., Akhtar, M. F., Saleem, A., Khalifa, S. M. A., El-Seedi, H. R., Kamel, M., Albadrani, G. M., Abdel-Daim, M. M., & Mittal, V. (2021). Acetylcholinesterase Inhibitory Potential of Various Sesquiterpene Analogues for Alzheimer’s Disease Therapy. In Biomolecules (Vol. 11, Issue 3, p. 350). https://doi.org/10.3390/biom11030350.

Materials and methods:

  1. Could you explain briefly the extraction procedure.
  2. Ethyl acetate is slightly less polar than chloroform. Why did you use both of them?
  3. How did you evaporate the solvent?
  4. Which solvent was used to dissolve the extract and fill the column space?
  5. What are the parameters you used when collecting the separated fractions?
  6. Line 89, please correct “essay”.
  7. For each assay, clarify if it was conducted on the extract or the fraction.
  8. Is it rational to analyze the aqueous fraction using GC-MS?
  9. Please, make sure that you cite an appropriate reference in each experiment.
  10. “using elevated plus maze, light/dark test and hole board test”, please explain in more details.
  11. Please explain briefly the biological assays of the manuscript.
  12. The authors could benefit from the following reference in isolation and fractionation procedures:

Zahra, M.H., Salem, T.A., El-Aarag, B., Yosri, N., El-Ghlban, S., Zaki, K., Marei, A.H., El-Wahed, A., Saeed, A., Khatib, A. and AlAjmi, M.F., 2019. Alpinia zerumbet (Pers.): Food and medicinal plant with potential in vitro and in vivo anti-cancer activities. Molecules, 24(13), p.2495.

Results and discussion:

  1. Better move figure 1 to the materials and methods part.
  2. How is the first solvent ratio is 25:50 in figure 1?
  3. Could you double-check the three last solvent systems?
  4. Could you add the charts of the identified metabolites made by GC-MS?
  5. Is silane one of the plant metabolites?
  6. what about the bioactivity of the other identified metabolites?
  7. Expand on the conclusion part and add the future perspectives.

Author Response

(The authors gave the same response as above.)

Reviewer 4 Report

Dear Authors

The present paper “Memory enhancing potential of Lavandula stoechas L.: Evidence-based approach”is very interesting and innovative;moreover, it is very smooth and pleasant to read.I have few objections to this work and ask you for some clarifications.

1) Do you think that plants belonging to species other than Lavandula stoechas may have similar biological properties?

2) Have you measured the antioxidant properties and polyphenolic content of the extract in vitro? If so, please indicate the results in the work

3) Reagents not used in the work are indicated in Chemicals and Drugs.

Please check correspondence. 

Author Response

(The authors gave the same response as above.)

Round 2

Reviewer 2 Report

The Authors have corrected the manuscript. I have no further comments.

Author Response

Dear Sir

It is stated that I want to submit revised article entitled, “Biomolecular Evaluation of Lavandula stoechas L. for Nootropic Activity” for publication in your esteemed Journal. We are highly thankful to referees whose comments helped in improving this manuscript. We have revised the entire manuscript for language as well as for proper flow of the information. Mostly these are the reviewers’ observations which are addressed in the point by point rebuttal file and also incorporated the same in the text. Below is response to referee comments:

Reviewer Comment: Again "Evidence-based approach" is not suitable for the title; please explain what is your evidence? Could you add evaluation or assessment

Authors response: Corrected New Title is, “Biomolecular Evaluation  of Lavandula stoechas L. for Nootropic Activity”

Reviewer Comment: The graphical abstract is highly recommended.; I don`t see it in the second version

Authors response: Provided

Reviewer Comment: " Phenethylamine and α-tocopherol were detected in AfL.s by using GC-MS"; do you mean Phenethylamine and α-tocopherol are the major components?

Authors response: After purification of extract of L. stoechs, we obtained different fractions, all the fractions were tested for inhibition of AChE activity. One fraction out of those which produced maximum inhibition of AChE activity was then subjected to GC-MS for the identification of biomolecules. The results of GC-MS analysis indicated the presence of two major comonents i.e. " Phenethylamine and α-tocopherol. That fraction was then used for Behavioral and Biochemical studies. The results of  Behavioral and Biochemical studies proved that both of these were responsible for enhancement of memory. In literature, Phenethylamine has been mentioned as memory enhancer and α-tocopherol as potent anti osidant. The references are given in discussion.

Reviewer Comment: The authors would add the originality and the novelty of their work.

Authors response: Added in the discussion of the manuscript

Reviewer Comment: " column chromatography" which type of CC?

Authors response: Column chromatography was used for the separation of different constituents of the aqueous extract of the Plant. Liquid chromatography was used in CC.

Reviewer Comment: Again GC-MS is used for Essential oil chemically characterizations so why the authors didn`t use other methods like NMR and MS-MS.

Authors response: We used GC-MS because of its availability in our institute. Moreover, NMR and MS-MS were not used because it was difficult to manage these testing due to lack of facilities.

Reviewer Comment: The abstract is confusing, please re-write?

Authors response: Rewritten

Reviewer Comment: Please add the purity percentage of the chemicals

Authors response: Provided

Reviewer Comment: Could you add positive control for all the biological assays

Authors response: We used simple UV-visible spectrophometer for enzyme assays. It was difficult to arrange the enzymes like AChE, Choline Acetyl transferase. Spectrophotometric method is mostly used for detection of various enzymes at this lower level.

Reviewer Comment: Please supply IC50 values for all biological assays

Authors response: The IC50 Value of methanolic extract of L.stoechas was found and is reported in previous published paper. It is not mentioned in this manuscript to avoid the plagiarism. The IC50 Values of other assays were not found.

Graphical Abstract - in attachment

Thank you once again for your valuable comments. I am available if there are any further queries.

--

Best regards,

Reviewer 3 Report

May 29, 2021

Journal: Plants

Title: Memory enhancing potential of Lavandula stoechas L.: Evidence-based approach

Authors: Aamir Mushtaq, Rukhsana Anwar, Umar Farooq Gohar, Mobasher Ahmad, Marc Romina Alina, Rector Research, Marius Irimie, Elena Bobescu

Dear Editor:

The authors have investigated the potential of L. stoechas biomolecules for the management of dementia, in-vitro anti-cholinesterase activity, as well as in-vivo (behavioral and biochemical) studies. The manuscript could be accepted after major revision.

Hesham El-Seedi, professor,
Associate Editor

Journal of Advanced Research, I.F.= 6.99

--------------------------------------

Department of Molecular Biosciences,

The Wenner-Gren Institute,

Stockholm University, S-106 91,

Stockholm, Sweden

Tel: +46-700 43 43 43

E-mail address: hesham.elseedi@su.se

-----------------------------------------

Pharmacognosy Division,

BMC, Uppsala University,

SE-751 23 Uppsala, Sweden

Email: hesham.el-seedi@farmbio.uu.se

Comments to authors:

  1. Again " Evidence-based approach" is not suitable for the title; please explain what is your evidence? Could you add evaluation or assessment
  2. The graphical abstract is highly recommended.; I don`t see it in the second version
  3. " Phenethylamine and α-tocopherol were detected in AfL.s by using GC-MS"; do you mean Phenethylamine and α-tocopherol are the major components?
  4. The authors would add the originality and the novelty of their work.
  5. " column chromatography" which type of CC?
  6. Again GC-MS is used for Essential oil chemically characterizations so why the authors didn`t use other methods like NMR and MS-MS
  7. The abstract is confusing, please re-write?
  8. Please add the purity percentage of the chemicals
  9. Could you add positive control for all the biological assays
  10. Please supply IC50 values for all biological assays

Author Response

(The authors gave the same response as above.)
